# Etiologies and Trends in Extremity Amputations: A Ten-Year Single-Center Experience

**DOI:** 10.3390/healthcare13182256

**Published:** 2025-09-09

**Authors:** Abdulrahman Alaseem, Mishari Alanezi, Mohammed N. Alhuqbani, Zyad A. Aldosari, Faisal Alkhunein, Khalid Alyahya, Khalid Alanezi, Mohammad N. Aljarba, Musaad Alhamzah, Ibrahim Alshaygy, Waleed Albishi

**Affiliations:** 1Department of Orthopedic Surgery, College of Medicine, King Saud University, Riyadh 12372, Saudi Arabia; 2College of Medicine, King Saud University, Riyadh 12372, Saudi Arabia; 3Department of Surgery, College of Medicine, King Saud University, Riyadh 12372, Saudi Arabia

**Keywords:** limb amputation, diabetic foot, public health, epidemiology

## Abstract

**Background**: Limb amputation is a life-altering event with substantial physical, psychological, and social consequences. Despite advances in healthcare, amputation remains a major global health concern, particularly in regions with high burdens of diabetes and vascular disease. This study aims to analyze demographic characteristics, etiologies, and trends in amputations over a ten-year period at a tertiary-care center. **Methods**: We retrospectively reviewed medical records of patients who underwent amputation at our tertiary-care hospital. Collected variables included patients’ demographics, etiology, level of amputation, type of admission, and surgical specialty involved. Descriptive statistics were used to summarize the data. Associations between categorical variables were analyzed using chi-square tests, with post hoc pairwise comparisons adjusted using the Bonferroni method where applicable. Continuous variables were compared using the Mann–Whitney U test. A *p*-value of <0.05 was considered statistically significant. All analyses were conducted using IBM SPSS version 26.0. **Results**: A total of 647 patients underwent amputation, with a mean age of 56 years and a male predominance (65%). Diabetic complications were the leading cause (67.7%), followed by trauma (11.7%) and vascular diseases (11.6%). Lower limb amputations were more prevalent, with toe amputations being the most frequent (39%). Emergency procedures accounted for 72% of cases, and vascular surgery was the most involved specialty, followed by orthopedic surgery. **Conclusions**: Our study highlights a substantial burden of amputations, predominantly involving the lower limb, as well as a significant association with diabetic complications. These findings emphasize the urgent need for integrated diabetic care, early interventions, and public health strategies to reduce the burden of amputations in Saudi Arabia.

## 1. Introduction

Amputation, the surgical removal of all or part of a limb, is a life-changing procedure with profound physical, psychological, and social consequences [1]. It is a major preventable public health concern that can substantially affect patients’ mobility, independence, and quality of life [2]. Despite advancements in healthcare and in limb-salvage surgical techniques, amputations remain prevalent worldwide. In the United States (US), an estimated 1.6 million individuals are living with a lost limb, a number expected to double by 2050. Furthermore, approximately 185,000 amputations occur annually in the US alone [3]. Similarly, countries in the Middle East are witnessing a rising burden of non-communicable diseases, particularly diabetes mellitus, which has become a major contributor to limb amputation [4].

The epidemiology of amputations varies considerably across regions, influenced by differences in healthcare access, preventive services, and disease burden [5]. Globally, the most common indications for amputation include diabetic foot complications, ischemic vascular disease, and traumatic injuries [6]. Vascular diseases—in particular, peripheral arterial disease (PAD)—along with diabetic foot complications account for more than 90% of lower limb amputations in high-income countries [5]. In contrast, trauma-related amputations are more prevalent in low- and middle-income countries [7]. Over the past decades, trends in amputations have fluctuated greatly due to improvements in diabetes management, revascularization techniques, and preventive strategies, including earlier diagnosis of PAD and better glycemic control in diabetic patients [8]. However, despite these improvements, the overall burden of amputations remains substantially high [9]. In many regions, the rising prevalence of diabetes mellitus and peripheral vascular disease contributes greatly to the increased incidence of lower limb amputation [10]. Moreover, the limited healthcare access and higher rate of traumatic injuries in certain regions further contribute to the persistently high amputation rate [11].

Data regarding prevalence, etiologies, and trends in amputations are limited due to the lack of a national amputation registry [4]. Existing studies suggest that trauma has been historically the leading cause of amputation, although more recent trends remain unclear [12]. Understanding the patterns and etiologies of amputation is crucial for developing targeted preventive strategies, improving rehabilitation services, and optimizing patient outcomes. Our study aims to address this gap by presenting a decade-long retrospective analysis of limb amputations at our high-volume tertiary health institution, providing valuable insight for healthcare policymakers in developing preventive strategies and improving patient outcomes.

## 2. Materials and Methods

### 2.1. Study Design, Setting, and Population

This retrospective cohort study was conducted at King Saud University Medical City, a tertiary-care and referral center in Riyadh, Saudi Arabia. The institution functions as a high-volume center and receives referrals for complex trauma, diabetic foot disease, vascular disorders, oncologic resections, and reconstructive procedures from across the country. We reviewed all consecutive cases of upper and lower limb amputations performed between January 2015 and December 2024. Patients of all age groups and both genders were eligible for inclusion, while those with incomplete medical record documentation were excluded.

The study was approved by the Institutional Review Board (IRB) of the College of Medicine, King Saud University (Project No. E-24-9329). Given the retrospective nature of the study, the IRB waived the requirement for informed consent.

### 2.2. Data Collection and Variables

Data were extracted from Electronic Medical Records (EMRs) and recorded in a computerized database. Collected variables included demographic characteristics (age, gender, marital status, and nationality) and amputation-related details such as etiology, level of amputation, admission type (emergency vs. elective), surgical specialty performing the procedure, and date of surgery. Etiology of amputations was classified into six categories: diabetic complications, including patients with documented diabetes mellitus who underwent amputation for diabetic foot ulcers, gangrene, or infection, with diabetes considered the primary underlying cause of the amputation; vascular diseases, including only non-diabetic patients with peripheral arterial disease, critical limb ischemia, or other vascular pathologies; trauma, referring to amputations resulting from acute injuries caused by high-energy mechanisms (e.g., motor vehicle accidents, industrial or work-related crush injuries) or domestic incidents (e.g., machinery accidents, heavy-object injuries, penetrating trauma); malignancy, including amputations performed for primary bone or soft tissue sarcomas, metastatic tumors, or other malignant neoplasms requiring amputation for oncologic control; deformities, involving severe congenital or acquired conditions such as contractures or rotational anomalies that are not amenable to reconstructive or limb-salvage procedures; and infections, defined as non-diabetic soft tissue or bone infections unresponsive to medical therapy and conservative surgical intervention. 

Amputations were performed by four surgical specialties: vascular surgery, orthopedic surgery, plastic surgery, and general surgery.

### 2.3. Statistical Analysis

Descriptive statistics were used to summarize the demographic and clinical characteristics of the study population. Frequencies and percentages were used for categorical variables, while continuous variables were summarized using means and standard deviations (SD). Associations between categorical variables were assessed using chi-square tests, with post hoc pairwise comparisons adjusted using the Bonferroni method where applicable. For continuous variables, comparisons were performed using the Mann–Whitney U test due to non-normal distribution. A *p*-value of <0.05 was considered statistically significant for all analyses. All statistical analyses were conducted using IBM SPSS Statistics version 26.0 (IBM Corporation, Armonk, NY, USA).

## 3. Results

### 3.1. Demographic Characteristics

Over the past decade, 647 patients underwent amputation, with no cases excluded from the analysis. Of these, 543 patients (84%) required a single amputation, while 104 patients (16%) underwent multiple amputations, resulting in a total of 769 amputation procedures performed. The mean age of the population was 56.1 ± 21.1 years (range: 1–98 years). Most patients were male (65%), with the highest proportion of amputations occurring in those aged between 60 and 79 years (41.9%), followed by the 40–59 age group (31.7%). The youngest (0–19 years) and the oldest (80 and older) age groups accounted for 9% and 9.7% of the cases, respectively. The distribution of age groups showed a statistically significant difference (*p* < 0.05) (Table 1).

### 3.2. Annual Trends in Amputations

The number of amputations varied over the years, with an overall rising trend observed throughout the study period. The annual number of cases increased from 28 cases in 2015 to a peak of 95 cases in 2022. A slight decline from this peak was noted in 2023 (76 cases) and 2024 (87 cases). Furthermore, there was a notable dip in 2020 to 48 cases only, most likely related to pandemic-related healthcare disruption (Figure 1). The overall trend indicates a rising burden of amputations, emphasizing the need for improved preventive measures and management strategies to address the underlying causes of limb loss.

### 3.3. Etiologies of Amputation, and Reamputation Rates

Regarding etiology, diabetic complications were the leading cause of amputation (67.7%, *n* = 438; *p* < 0.05) over the past 10 years, highlighting the substantial burden of diabetes-related limb loss. Other causes included trauma (11.7%), vascular diseases (11.6%), infection (4%), deformities (3%), and malignancy (2%). Furthermore, the distribution of etiologies across age groups varied significantly (*p* < 0.05). Notably, the younger population (0–19 years) demonstrated a distinct profile, with trauma being the predominant cause (69%), followed by deformities (24%) and malignancy (7%), with no cases attributed to diabetic complications, vascular diseases, or infection. Diabetic complications were most prevalent in the 40–79 age range (82% in 40–59, 81% in 60–79), while trauma accounted for 50% in those aged 20–39, dropping to 4% in the 40–59 group, and becoming nearly absent in individuals aged 80 years or older. In contrast, vascular-related amputations increased with age, starting at 0% in individuals under 20, rising to 10% in the 20–39 group, and peaking at 33% in those aged 80 and above. Infections and malignancies remained low across all age groups (Figure 2). Post hoc analysis with Bonferroni correction showed that trauma was significantly more frequent in the 0–19 years (*p* < 0.001) and 20–39 years (*p* < 0.001) groups, and markedly less common in the 60–79 years group (*p* < 0.001). Deformities were more frequent in the 0–19 years group (*p* < 0.001), whereas vascular causes were more prevalent among those aged ≥ 80 years (*p* < 0.001). Diabetic complications, although initially appearing underrepresented in younger groups, did not retain statistical significance after Bonferroni correction, indicating their age distribution was not significantly different from expected. Similarly, malignancy and infection did not show significant deviations across age categories.

Moreover, the distribution of etiologies across genders was not statistically significant, as diabetic complications, malignancy, and infections were equally distributed in both males and females. However, trauma was more common in males, while vascular and congenital causes were slightly higher in females (Figure 3). Among traumatic amputations (*n* = 76, 11.7%), the majority were caused by motor vehicle accidents (58%), followed by industrial/work-related crush injuries (25%) and domestic accidents (17%). High-energy trauma (motor vehicle accidents and industrial injuries) was most frequently associated with lower limb amputations, while low-energy mechanisms such as domestic crush injuries and sharp-force trauma were more commonly associated with upper limb amputations. Regarding reamputation cases (*n* = 104), diabetic complications were the leading cause (90%), followed by vascular causes (7%) and infections (3%). Notably, no traumatic amputations required reamputation. Most reamputations were elective (67%), while 33% were performed as emergency procedures.

Among Saudi nationals (*n* = 546), diabetic complications accounted for 69.6% of amputations, followed by vascular causes (11.0%) and trauma (9.5%). In contrast, among non-Saudi nationals (*n* = 101), the proportion of diabetic complications was lower (57.4%), with trauma (23.8%) and vascular-related amputations (14.9%) being more prevalent, these differences being statistically significant (*p* < 0.05).

### 3.4. Level of Amputation

Regarding the level of amputation, lower limb amputations were more common than upper limb amputations, with toe amputations accounting for 39% (n = 254), followed by below-knee amputations at 25% (*n* = 161) and above-knee amputations at 20% (n = 128). Less frequently performed procedures included metatarsal (5%), Chopart (1%), and, in rare cases, Lisfranc, Syme, and hip disarticulations (<1%). This distribution was statistically significant (*p* < 0.05), indicating a predominance of distal limb involvement, particularly toe amputations. Moreover, upper limb amputations were less frequent, with finger amputations being most common (8%), followed by above-elbow (1%) and below-elbow (0.5%). Other procedures, including forequarter and wrist amputations, were rare, with statistically significant variation being observed (*p* < 0.05). Notably, trauma was the most common cause of upper limb amputations, whereas diabetic complications were the predominant cause of lower limb amputations. Additionally, the distribution of amputation levels varied notably across age groups. In the younger population (0–19 years), toe (40%) and finger amputations (40%) predominated, while major amputations were rare (above-knee 2%, below-knee 7%). In contrast, older adults demonstrated a higher prevalence of major lower limb amputations. Below-knee amputations peaked at 29% in the 40–59 age group, whereas in the 60–79 group, toe amputations were most common (45%), followed by below-knee (27%) and above-knee (22%) amputations. In the oldest group (≥80 years), above-knee amputations predominated (40%).

### 3.5. Type of Admission and Surgical Specialty Involved

Most amputations were performed as emergency cases (*n* = 463, 72%), while 181 patients (28%) underwent planned, elective amputations. The significantly higher rate of emergency amputations (*p* < 0.05) highlights the acute nature of conditions necessitating urgent surgical intervention (Table 2). Emergency amputations were most often associated with diabetic complications (73%) and trauma (14%), with vascular causes (9%) and infections (4%) being less common. Elective amputations showed a distinct profile, with diabetic complications accounting for 55%, followed by vascular disease (18%), deformities (9%), malignancy (7%), trauma (7%), and infections (4%).

Regarding surgical specialty involvement, vascular surgeons performed the highest number of amputations (*n* = 316, 48%), followed by orthopedic surgeons (*n* = 200, 30%). Plastic surgeons (*n* = 79, 12%) and general surgeons (*n* = 52, 8%) accounted for smaller proportions of cases. A significant distribution across specialties was recorded (*p* < 0.05).

## 4. Discussion

This retrospective study analyzed the demographic trends, etiologies, and anatomical distribution of amputations over a 10-year period. The findings of our study highlight a substantial burden of amputations, with a clear predominance of lower limb involvement and a significant association with diabetic complications being recorded. These results are largely consistent with international studies [3,13] and offer insights that can improve preventive measures and patient outcomes. 

Globally, limb amputation remains a major cause of long-term disability, with considerable variation in incidence across regions due to differences in healthcare infrastructure, disease burden, and injury patterns. In the Middle East, the increasing prevalence of non-communicable diseases, driven by rapid urbanization, dietary changes, and sedentary lifestyles, has amplified the burden of amputation [4]. Nevertheless, the lack of comprehensive studies assessing the trends of amputation limits the ability of healthcare policymakers to develop targeted preventive strategies to reduce the overall burden of amputation. To address this gap, our study provides a decade-long retrospective analysis of extremity amputations, aiming to identify their underlying etiologies, temporal trends, and demographic patterns within our population.

Our findings revealed a significant male predominance, with males comprising 65% of the study population. This finding is consistent with international studies such as that of Ziegler-Graham et al. [3], who reported that men are more likely to undergo limb amputation than women. This gender variation may be attributed to the greater exposure to trauma, as well as the higher prevalence of diabetes and peripheral vascular disease, among men [13]. Alshehri et al. [1] reported similar trends, particularly in trauma-related amputation, where men are disproportionately affected, reinforcing the idea that occupational factors contribute to this disparity.

The mean age of our patients was 56 years, with peak incidence in the 60–79 age group. This age distribution is consistent with Shahine et al. [12], who found that older adults with multiple comorbidities, especially diabetes and vascular diseases, were more frequently affected. Moreover, our study showed that 9% of amputations occurred in the 0–19 age group, in line with trends reported in the literature for Western countries [14], highlighting the prevalence of limb loss among younger population groups. 

Diabetic complications emerged as the leading cause, accounting for 67.7% of all amputations, comparable to the 68.6% reported in Poland by Walicka et al. [15]. Furthermore, Almohammadi et al. [16] reported that 84.9% of patients with diabetic foot eventually undergo amputation. The increasing trend of diabetic amputations likely reflects challenges such as inadequate preventive strategies, poor glycemic control, and late presentation. 

Trauma was the second most frequent etiology (11.7%), disproportionately affecting younger males. This is consistent with the findings of Dhillon et al. [17], who reported a mean age of 35.6 years among trauma-related amputees. Similarly, Yuan et al. [9] reported that traumatic amputations remain common among younger individuals, particularly in high-risk settings. 

Traumatic amputations accounted for 69% in those aged 0–19, and 50% in the 20–39 age group. In contrast, vascular causes were found to increase with age, peaking at 33% among individuals aged 80 and above, highlighting an age-related shift in the etiology of amputation. This trend aligns with Shahine et al. [12], who reported that trauma accounted for 69.2% of amputations in younger age groups, whereas vascular-related amputations peaked at 89.5% in patients aged 70 and above. These findings support the observed age-related changes in the etiology of amputation.

The distribution of trauma versus vascular indications for amputations varies greatly according to the country’s level of industrial development [1]. In lower-income countries, trauma remains a leading cause, largely due to limited occupational safety measures and higher exposure to accidents [7]. In contrast, in more industrialized nations, vascular causes are more prevalent, reflecting lifestyle-related health burdens and aging populations [5].

From an anatomical standpoint, lower limb amputations were far more prevalent, with toes (39%), below-knee (25%), and above-knee (20%) being the most common. This distal-to-proximal progression reflects the natural course of diabetic foot disease [18,19]. Upper limb amputations were less frequent, and mostly associated with trauma, which aligns with the distribution patterns described by Yuan et al. [9].

In our study, most amputations were performed as emergency procedures (72%), compared to only 28% being planned, elective amputations. This statistically significant difference highlights the acute clinical presentation of many patients requiring limb amputation. The predominance of emergency cases suggests late presentation, delayed referral, or rapid progression of underlying conditions. Similar patterns have been observed in other studies. For instance, Dualeh et al. [20] found that over 54% of amputations were performed as emergency procedures, and attributed this to poor healthcare access and inadequate preventive care. In contrast, Rice et al. [21] reported that more than 62% of amputations were performed electively, reflecting more-structured surgical planning, likely explained by routine surveillance and multidisciplinary patient care. The significantly higher rate of emergency amputations in our cohort indicates an urgent need to improve preventive strategies, as emergency amputations are typically associated with higher postoperative complications, longer hospital stays, and worse rehabilitation outcomes [21].

From a policy and public healthcare perspective, our findings stress the urgent need for earlier diabetes screening, better glycemic control programs, and integrated diabetic foot care services. Establishing screening clinics, especially for high-risk diabetic patients, could play a transformative role in detecting complications early and reducing emergency procedures. A systematic review reported that regular foot screening in individuals with diabetes may lead to a reduction in major amputations by up to 96%, highlighting the potential of early detection and intervention [22]. Additionally, a study in Singapore found that diabetic patients who underwent regular foot screening had significantly lower rates of major and minor amputations compared to those who did not receive screening [23]. These findings emphasize the effectiveness of routine foot examinations in preventing amputation.

The strength of this study lies in its extended ten-year timeframe and large sample size, which enabled a comprehensive analysis of evolving trends across age groups and etiologies. However, several limitations must be acknowledged. First, the use of single-center data may affect the generalizability of the findings to other healthcare settings, even though our institution is one of the largest tertiary-care centers in Saudi Arabia and receives referrals across the country. Second, the retrospective nature of the study introduces potential biases due to incomplete medical records documentation. Third, the absence of long-term outcomes, such as mortality, rehabilitation progress, and quality of life, limits our ability to assess the full impact of amputation on patient well-being. Future studies should aim to develop multicenter amputation registries to capture nationwide epidemiological trends and facilitate benchmarking across institutions. Prospective cohort studies assessing the long-term outcomes of amputees, including functional recovery, prosthesis use, quality of life, and healthcare costs, would provide valuable insights for patient-centered care planning. Interventional trials evaluating structured diabetic foot care programs, occupational injury prevention strategies, and community-based screening initiatives could directly inform health policy and reduce the incidence of amputation.

## 5. Conclusions

In conclusion, this study provides valuable insights into epidemiological and etiological trends in limb amputations over a ten-year period. The findings highlight the predominant role of diabetic complications as the leading cause of amputation, the high prevalence of lower limb involvement, and the significant proportion of emergency procedures. These results highlight the importance of early diabetic management and enhanced preventive care to improve overall patient outcomes. To reduce the incidence of emergency amputation, national strategies should include developing standardized screening guidelines along with targeted training for primary-care physicians in the early detection of limb-threatening conditions.

## Figures and Tables

**Figure 1 healthcare-13-02256-f001:**
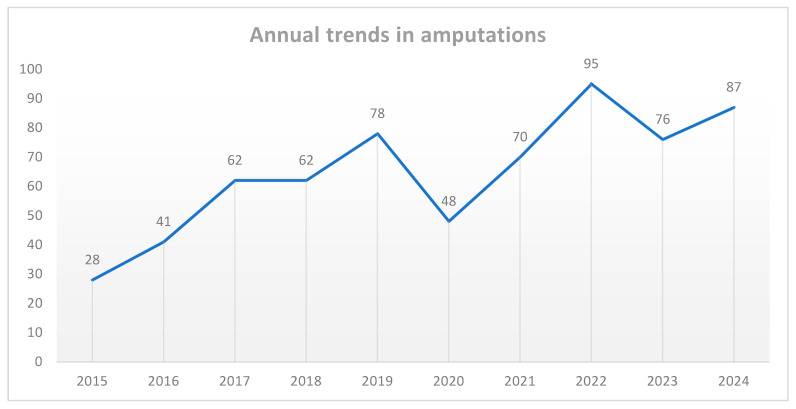
Annual trends in amputations during the study period.

**Figure 2 healthcare-13-02256-f002:**
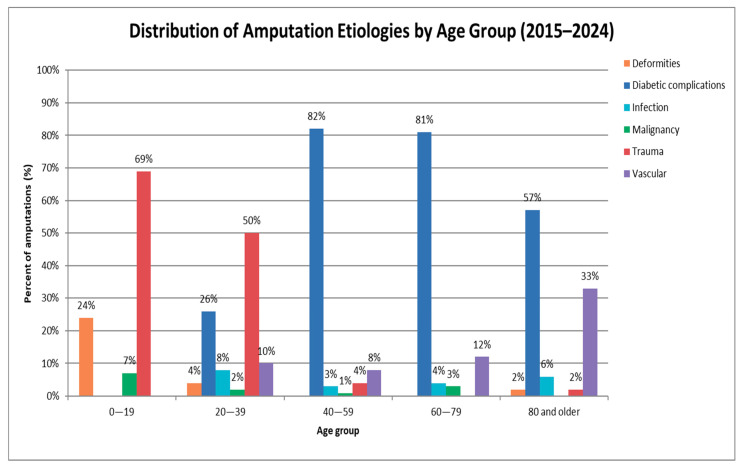
Distribution of etiologies across age groups.

**Figure 3 healthcare-13-02256-f003:**
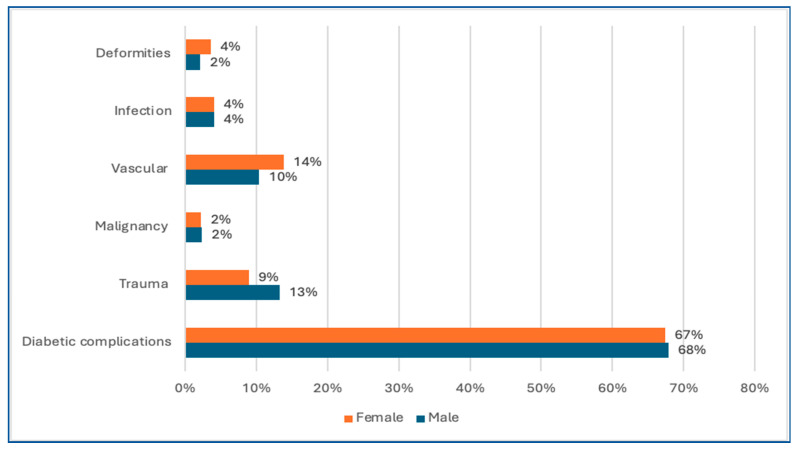
Distribution of etiologies across gender.

**Table 1 healthcare-13-02256-t001:** Demographic characteristics of patients undergoing amputations (2015–2024).

Demographics		Value	Percentage	*p* Value
Total patients	*n*	647	-	-
Age	Whole population (mean SD)	56.12 (21.1)	-	-
	Male age (mean SD)	55.47 (20.2)	-	0.03115
	Female age (mean SD)	57.34 (22.7)	-	
Gender	Male	423	65.4%	<0.05
	Female	224	34.6%	
Nationality	Saudi	546	84.4%	<0.05
	Non-Saudi	101	15.6%	
Marital Status	Married	382	59%	<0.05
	Single	256	39.6%	
	Divorced	7	1.1%	
	Widowed	2	0.3%	
Age groups	0–19	58	9%	<0.05
	20–39	50	7.7%	
	40–59	205	31.7%	
	60–79	271	41.9%	
	80 and older	63	9.7%	

**Table 2 healthcare-13-02256-t002:** Etiology, level of amputation, admission type, and surgical specialty involved.

Variables	Category	*n*	%	*p* Value
Etiology	Diabetic complications	438	67.7%	<0.05
	Trauma	76	11.7%	
	Vascular diseases	75	11.6%	
	Infection	26	4.0%	
	Deformities	17	2.6%	
	Malignancy	15	2.3%	
Distribution of amputation levels (Lower limb)	Above-knee Amputation	128	19.8%	<0.05
	Below-knee Amputation	160	24.7%	
	Toe Amputation	254	39.3%	
	Metatarsal Amputation	30	4.6%	
	Lisfranc Amputation	3	0.5%	
	Syme Amputation	2	0.3%	
	Chopart Amputation	5	0.8%	
	Hip Disarticulation	1	0.2%	
Distribution of amputation levels (Upper limb)	Forequarter Amputation	1	0.2%	<0.05
	Above-elbow Amputation	8	1.2%	
	Below-elbow Amputation	3	0.5%	
	Wrist Amputation	3	0.5%	
	Finger Amputation	49	7.6%	
Type of admission	Emergency	466	72.0%	<0.05
	Elective	181	28.0%	
Surgical Specialty	Vascular Surgery	316	48.8%	<0.05
	Orthopedic Surgery	200	30.9%	
	Plastic Surgery	79	12.2%	
	General Surgery	52	8.0%	

## Data Availability

The dataset used and analyzed in this study is available from the corresponding author.

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
