# Peer review of "Etiologies and Trends in Extremity Amputations: A Ten-Year Single-Center Experience"

_healthcare, 2025, doi:10.3390/healthcare13182256_

Round 1

Reviewer 1 Report

Comments and Suggestions for Authors

I congratulate Alanezi et al with their study healthcare-3803895 “Etiologies and Trends of Extremities Amputations: A Ten-Year 2 Single Center Experience”. This is interesting manuscript aiming to describe authors experience in managing patients who require amputations.
At revision I have identified multiple issues that should be addressed at revision to improve the manuscript
1. The hypothesis of the study is a bit unclear. Analyses of your data reveal that you actually have two groups of the patients with diabetes (n = 438) and without diabetes (n= 210). In this matter, I think it would be better to reconsider you data within these two groups. Otherwise, one is confused by various levels of amputations, age etc. It is more interesting to show analyses of diabetes vs non-diabetes related amputations because of different ethologies and outcomes. In such approach it is also easier to apply statistical analyses for tow groups, rather than split the data for > 3 groups as I see from you paper (Table 1 and 2).
2. In case Authors want keep their data presentation in the current shape, they need to do Statistical analysis with more explanations and probably recalculations. Authors should explain what kind of statistical tests have been applied for data in Table 1 and 2? There are more than 2 groups to be presented, which mean that you should use non-parametrical tests for multiple groups (e.g. Kruskal–Wallis test). Having 2 groups comparisons it is good to apply Mann–Whitney U test.
3. I disagree with Authors to use - Student t test, because in medical studies one can not expect normal distribution in the variable, therefore non-parametrical analyses must be applied.
4. Authors should also add a data about the reamputations rate, especially for the traumatic amputations cases. If you had a reamputation, please, add more details about those cases.

Reviewer 2 Report

Comments and Suggestions for Authors

In the above-mentioned paper, the authors presented the epidemiology of amputations at their institution over a ten-year period. Overall, the manuscript is clear and well-structured, and the authors adhered to the journal’s submission guidelines. However, the study reports only basic epidemiological data, omitting key parameters such as long-term outcomes, mortality, and quality of life. These were acknowledged by the authors as limitations, and their absence significantly reduces the scientific value of the paper. Given that trauma was analyzed as a distinct cause of amputation, it is necessary to provide a more detailed description of the most common types of traumatic injuries. The authors also stated that all patients were included in the analysis regardless of age; however, it is essential to present specific results for patients under the age of 18. Following revision, the manuscript will be reconsidered for publication.

Reviewer 3 Report

Comments and Suggestions for Authors

Thank you for this interesting article. The sample size is considerable and offers a valuable opportunity to conduct more in-depth statistical analyses. I would be genuinely interested in seeing more details (see comments provided).

Suggestions:

Lines 86–87: Given that the categories "diabetic complications" and "vascular diseases" are not exactly mutually exclusive, could you be more specific about how patients were assigned to each category depending on the different situations encountered?

Lines 95 to 98: Examples of relationships analyzed do not need to be provided in the Materials and Methods section, or alternatively, the list of statistical relationships explored should be exhaustive.

Figure 1: It is very interesting to present this information graphically. It might also be useful to show the demographic evolution of the population in Saudi Arabia over the same period.

Line 103: How many patients were excluded due to missing medical data or incomplete reports?

Line 127: Were any post-hoc analyses performed to determine between which specific groups the significant relationship was observed? Would it be possible to include a graph showing the distribution of etiologies across the different age groups (as was done for gender)?

Lines 142–145: Were these differences statistically significant, or do they rather represent a trend?

Lines 159–162: What are the proportions of the different etiologies between emergency cases and elective amputations?

Lines 166–169: This is an interpretation and should rather be included in the Discussion section, not in the Results.

Lines 220–232: It would be interesting to better understand the differences between emergency and planned procedures, for example in terms of etiology.

Round 2

Reviewer 2 Report

Comments and Suggestions for Authors

Paper can be accepted in this form